# *iNOS* Gene Ablation Prevents Liver Fibrosis in Leptin-Deficient *ob/ob* Mice

**DOI:** 10.3390/genes10030184

**Published:** 2019-02-27

**Authors:** Sara Becerril, Amaia Rodríguez, Victoria Catalán, Beatriz Ramírez, Xabier Unamuno, Javier Gómez-Ambrosi, Gema Frühbeck

**Affiliations:** 1Metabolic Research Laboratory, Clínica Universidad de Navarra, 31009 Pamplona, Spain; arodmur@unav.es (A.R.); vcatalan@unav.es (V.C.); bearamirez@unav.es (B.R.); xunamuno@unav.es (X.U.); jagomez@unav.es (J.G.-A.); gfruhbeck@unav.es (G.F.); 2CIBER Fisiopatología de la Obesidad y Nutrición (CIBEROBN), Instituto de Salud Carlos III, 28029 Madrid, Spain; 3Obesity and Adipobiology Group, Instituto de Investigación Sanitaria de Navarra (IdiSNA), 31009 Pamplona, Spain; 4Medical Engineering Laboratory, University of Navarra, 31009 Pamplona, Spain; 5Department of Endocrinology & Nutrition, Clínica Universidad de Navarra, 31009 Pamplona, Spain

**Keywords:** Leptin, iNOS, Tenascin C, liver fibrosis, inflammation

## Abstract

The role of extracellular matrix (ECM) remodeling in fibrosis progression in nonalcoholic fatty liver disease (NAFLD) is complex and dynamic, involving the synthesis and degradation of different ECM components, including tenascin C (TNC). The aim was to analyze the influence of inducible nitric oxide synthase (*iNOS*) deletion on inflammation and ECM remodeling in the liver of *ob/ob* mice, since a functional relationship between leptin and iNOS has been described. The expression of molecules involved in inflammation and ECM remodeling was analyzed in the liver of double knockout (DBKO) mice simultaneously lacking the *ob* and the *iNOS* genes. Moreover, the effect of leptin was studied in the livers of *ob/ob* mice and compared to wild-type rodents. Liver inflammation and fibrosis were increased in leptin-deficient mice. As expected, leptin treatment reverted the obesity phenotype. *iNOS* deletion in *ob/ob* mice improved insulin sensitivity, inflammation, and fibrogenesis, as evidenced by lower macrophage infiltration and collagen deposition as well as downregulation of the proinflammatory and profibrogenic genes including *Tnc*. Circulating TNC levels were also decreased. Furthermore, leptin upregulated TNC expression and release via NO-dependent mechanisms in AML12 hepatic cells. *iNOS* deficiency in *ob/ob* mice improved liver inflammation and ECM remodeling-related genes, decreasing fibrosis, and metabolic dysfunction. The activation of iNOS by leptin is necessary for the synthesis and secretion of TNC in hepatocytes, suggesting an important role of this alarmin in the development of NAFLD.

## 1. Introduction

Obesity is associated with a wide spectrum of liver abnormalities, including nonalcoholic fatty liver disease (NAFLD), characterized by an increased intrahepatic triglyceride content. NAFLD is a major contributor to cardiovascular disease and a major cause of obesity-related morbidity and mortality [1]. It represents the most prominent form of liver diseases worldwide and ranges from simple fatty liver and non-alcoholic steatohepatitis (NASH) to liver cirrhosis and hepatocellular carcinoma [2]. Although the cause of NAFLD is unclear, a “multiple-hit” model of NAFLD development has been postulated considering multiple insults acting together on genetically predisposed subjects to develop NAFLD. Such hits include obesity, insulin resistance, hepatic lipid accumulation, activation of the inflammatory cascade, and fibrogenesis, as well as genetic and epigenetic factors [3]. Insulin resistance plays a key role, driving to an excessive de novo lipogenesis, a reduction of lipolysis, and the consequent increase of intrahepatic lipids [4]. Like other metabolic diseases, the generation of oxidative stress is another important contributor in the pathogenesis of NAFLD [5]. The excessive reactive oxygen species (ROS) participate in the development of NAFLD through different mechanisms, including lipid peroxidation and the subsequent activation of the nuclear transcription factor-κB (NF-κB) as well as the activation of hepatic stellate cells (HSC), the primary source of extracellular matrix (ECM) proteins such as collagen or tenascin C (TNC) [6]. NF-κB can bind to the inducible nitric oxide synthase (iNOS) promoter, inducing its expression and triggering the inflammatory process. The large amount of nitric oxide (NO) derived from iNOS stimulation acts in combination with ROS and produces nitrosative stress, creating a deleterious environment that leads to cell death and tissue damage [7].

Tenascin C is a multifunctional hexameric ECM glycoprotein undetectable in most healthy adult tissues, but highly expressed during embryonic development and dynamic tissue remodeling [8]. TNC modulates fibrotic and inflammatory responses in several diseases, including liver fibrosis, through the enhancement of the inflammatory response [9]. In this line, TNC is upregulated and deposited in both fibrotic areas and perisinusoidal spaces during liver diseases, with HSCs being considered its cellular source [10].

Leptin, the product of the *obese* (*ob*) gene, is also produced by HSCs after their activation [11]. In addition to its actions in the central nervous system, this adipocyte-derived hormone has direct effects on peripheral tissues, including the liver, the first adipokine directly associated with hepatic fibrosis [12]. Moreover, leptin mediates an inflammatory response by regulating the production of proinflammatory cytokines such as tumor necrosis factor alfa (TNF-α), interleukin-6 (IL-6), and IL-1β [13]. These cytokines also increase the secretion of leptin, sustaining a chronic proinflammatory state [14].

The functional relationship between leptin and iNOS has been described earlier by our group [15,16,17,18,19] and others [20,21]. In order to delve into the knowledge about the functional interplay between leptin and iNOS, which we previously demonstrated in double knockout (DBKO) mice lacking both genes [18,22], we hypothesized that the *iNOS* gene is involved in the liver inflammation and fibrosis development of *ob/ob* mice.

## 2. Material and Methods

### 2.1. Animals

The generation of a DBKO mouse simultaneously lacking the *ob* and the *iNOS* genes was performed as previously described [18]. Briefly, male *ob/ob* mice were intercrossed with female *iNOS* knockout mice (*iNOS*^-/-^) on a C57BL/6J background (Jackson Laboratories, Bar Harbor, ME, USA). Male mice were weaned at 21 days of age, genotyped, and maintained at room temperature (RT) on an artificial light–dark cycle (lights on from 8:00 a.m. to 8:00 p.m.) with a relative humidity of 50 ± 10% in a pathogen-free barrier facility. Mice had free access to water and were fed ad libitum a normal diet (ND) (12.1 kJ: 4% fat, 82% carbohydrate, and 14% protein, 2014S Teklad Global 14% Protein Rodent Maintenance Diet, Harlan, Barcelona). 

In a second subset of experiments, ten-week-old *ob/ob* mice were subclassified into three groups: control, leptin-treated (1 mg/kg/d), and pair-fed (*n* = 5 per group), as previously described [23]. The control and pair-fed groups received the vehicle (phosphate-buffered saline (PBS)), while leptin (Bachem, Bubendorf, Switzerland) was injected intraperitoneally twice a day at 8:00 a.m. and 8:00 p.m. for 28 days in the leptin-treated group. Control and leptin-treated groups were fed ad libitum with a standard chow diet (2014S Teklad, Harlan, Barcelona, Spain) [24], while the daily food intake of the pair-fed groups was matched to the amount eaten by the leptin-treated groups the day before to discriminate the inhibitory effect of leptin on appetite. 

Body weight and food intake were measured twice a week. The food efficiency ratio (FER) was determined as body weight gained per week divided by total energy (kcal) consumed over this period. 

Twelve-week-old mice were sacrificed by CO_2_ inhalation under fasting conditions. Sera samples were stored at −20 °C. Liver was carefully excised, weighed, frozen in liquid nitrogen, and stored at −80 °C. Hepatic biopsies were also fixed in 4% formaldehyde for immunohistochemical analyses. All experimental procedures conformed to the European Guidelines for the Care and Use of Laboratory Animals (Directive 2010/63/EU), and the study was approved by the Ethical Committee for Animal Experimentation of the University of Navarra. 

### 2.2. Blood Measurements

Blood samples were collected by cardiac puncture. Serum glucose was measured using a blood glucose meter (Ascensia Elite, Bayer, Barcelona, Spain). Concentrations of triglycerides, free fatty acids (FFA) (Wako Chemicals GmbH, Neuss, Germany), and glycerol (Sigma, St. Louis, MO, USA) were measured by enzymatic methods using commercially available kits as previously described [18]. Serum insulin and adiponectin were determined by ELISA (Crystal Chem Inc., Chicago, IL, USA and BioVendor Laboratory Medicine Inc., Modrice, Czech Republic, respectively). Intra- and inter-assay coefficients of variation for measurements of insulin and adiponectin were, respectively, 3.5 and 6.3% for the former, and 5.6 and 7.2% for the latter. The homeostatic model assessment (HOMA) was calculated as an indirect measure of insulin resistance with the formula: [fasting insulin (µU/mL) × fasting glucose (mmol/L)]/22.5. Circulating levels of TNC were determined by ELISA (IBL International GmbH, Hamburg, Germany). Intra- and inter-assay coefficients of variation for measurements of TNC were 3.5 and 6.3%, respectively.

### 2.3. Intrahepatic Lipid Content

The intrahepatic triglyceride content was measured by enzymatic methods, as previously described [25]. Briefly, tissues were homogenized and diluted in saline at a final concentration of 50 mg/mL. Homogenates were diluted (1:1) in 1% deoxycholate (Sigma) and incubated at 37 °C for 5 min. For triglyceride measurements, samples were diluted 1:100 in the reagent (Infinity Triglycerides Liquid Stable Reagent, Thermo Electron Corporation, Louisville, CO, USA) and incubated for 30 min at 37 °C. The resulting dye was measured based on its absorbance at 550 nm. Concentrations were determined compared with a standard curve of triglycerides (InfinityTM Triglycerides Standard, Thermo Electron). The protein content of the homogenates was measured by the Bradford method, using bovine serum albumin (BSA) (Sigma) as standard. All assays were carried out in duplicate.

### 2.4. Cell Cultures

A non-tumorigenic mouse hepatocyte cell line AML12 was purchased from American Type Culture Collection (Manassas, VA, USA) and maintained in a DMEM/F-12 medium (Invitrogen, Barcelona Spain) supplemented with 10% fetal bovine serum (FBS) (Invitrogen), 5 μg mL insulin, 5 μg mL transferrin, 5 ng mL selenium (Invitrogen), 40 ng m/L dexamethasone (Sigma), and antibiotic-antimycotic (complete growth medium). AML12 cells were plated 2 × 10^5^ cells/cm^2^ and grown in complete growth medium. AML12 hepatic cells were serum-starved for 24 h, and quiescent cells were stimulated with recombinant murine leptin (10 nmol/L) (450-31, PeproTech EC Inc., Rocky Hill, NJ, USA) in the presence or absence of L-N^6^-(1-iminoethyl)-lysine (L-NIL), a specific NOS inhibitor (10 µmol/L) (I8021, Sigma). Moreover, AML12 hepatocytes were serum-starved for 24 h and quiescent cells were stimulated with recombinant tenascin C (10 nmol/L) (3358-TC-050; R&D Systems, Minneapolis, MN, USA). The concentrations of leptin, tenascin C, and pharmacological inhibitor to perform the experiments were chosen on the basis of previous studies carried out by our group [22]. One sample per experiment was used to obtain control responses in the presence of the solvent.

### 2.5. RNA Extraction and Real-Time PCR

Total RNA was extracted from liver samples by homogenization with an ULTRA-TURRAX T 25 basic (IKA Werke GmbH, Staufen, Germany) using TRIzol Reagent (Invitrogen). RNA purification was carried out using the RNeasy Mini kit (Qiagen, Barcelona, Spain). All samples were treated with DNase (RNase-free DNase Set, Qiagen). The RNA concentration was determined from absorbance at 260 nm. For first-strand cDNA synthesis, constant amounts of total RNA were reverse transcribed using random hexamers as primers and M-MLV reverse transcriptase as previously described [26]. The transcript levels for *Tnc*, tumor necrosis factor-α (*Tnf*), toll-like receptor 4 (*Tlr4*), hypoxia inducible factor 1 alpha (*Hif1a*), CD11c (*Itgax*), *Cd44*, collagen type VI, *α*3 (*Col6a3*), collagen type VI, *α*1 (*Col6a1*), collagen type I, *α*1 (*Col1a1*), egf-like module-containing mucin-like hormone receptor-like 1 (*Emr1*), matrix metalloproteinase 9 (*Mmp9*), transforming growth factor β (*Tgfb*), α smooth muscle actin (*α*-SMA, *Acta2*), and osteopontin *(Spp1)* were quantified by real-time PCR (7300 Real Time PCR System, Applied Biosystems, Foster City, CA, USA). 

Primers and probes (Sigma) were designed using the software Primer Express 1.0 (Applied Biosystems) (Appendix A). The cDNA was amplified at the following conditions: 95 °C for 10 min, followed by 45 cycles of 15 s at 95 °C and for 1 min at 59 °C, using the TaqMan Universal PCR Master Mix (Applied Biosystems). The primer and probe concentrations for gene amplification were 300 and 200 nmol/L, respectively. All results were normalized to the levels of *18S* rRNA (Applied Biosystems), and the relative quantification was calculated using the ΔΔCt formula [27]. Relative messenger RNA (mRNA) expression was expressed as a fold expression over the calibrator sample. All samples were run in duplicate, and the average values were calculated.

### 2.6. Quantification and Characterization of Fibrotic Depots

Sections of formalin-fixed paraffin-embedded liver (6 µm) were dewaxed with xylene and rehydrated with decreasing concentrations of ethanol. Fibrosis was localized by Sirius Red staining (Sigma). Images of five fields per section from each animal were obtained at 200× magnification and the fibrous tissue area stained with Sirius Red/total amount of tissue was measured using the ImageJ analysis software, as described previously [28].

### 2.7. Statistical Analysis

Data are presented as the mean ± SEM. Differences between groups were assessed by unpaired two-tailed Student’s t-tests or two-way ANOVA as appropriate. In case of interaction between factors (lack of the *iNOS* or *ob* genes), a one-way ANOVA followed by Tukey’s or least significant difference (LSD) post hoc tests were applied. Moreover, comparisons between *ob/ob* groups and controls were analyzed by one-way ANOVA followed by Tukey’s post hoc tests. Statistics were calculated by the SPSS/Windows version 15.0 software (SPSS, Inc., Chicago, IL, USA). A *P*-value less than 0.05 was considered statistically significant.

## 3. Results

### 3.1. Lack of iNOS Gene Ameliorates the Obese Phenotype of *ob/ob* Mice

Anthropometric and metabolic variables of 12-week-old wild-type and leptin-deficient mice lacking the *iNOS* gene are shown in Table 1. As previously described [18,22], leptin-deficient *ob/ob* mice showed an increased (*P* < 0.001) body and liver weight that was significantly reduced in the absence of the *iNOS* gene. The absence of the *ob* gene was associated with insulin resistance, reflected by the increased (*P* < 0.001) levels of glucose, insulin, and HOMA index as well as by low adiponectin levels. Moreover, leptin deficiency was related with increased (*P* < 0.001) serum levels of FFA and glycerol. *iNOS* deficiency in *ob/ob* mice was associated with a significant improvement (*P* < 0.05) in glucose and lipid metabolism as compared to *ob/ob* mice counterparts. The analysis of intrahepatic TG content also revealed that *iNOS* deletion significantly (*P* < 0.05) decreased hepatosteatosis (Table 1).

### 3.2. *ob/ob* Mice Lacking iNOS Display Changes in the Expression of Molecules Involved in Liver Inflammation

The gene expression levels of key molecules involved in the proinflammatory response were analyzed in the liver of the experimental animals. The mRNA levels of the murine macrophage markers *Itgax* and *Cd68*, as well as *Tnf* and *Hif1a* were upregulated in *ob/ob* mice and significantly downregulated in *iNOS* deficient mice as compared to those of wild-type mice (Figure 1a–d; Appendix A). Furthermore, the gene expression levels of *Emr1*, *Itgax*, *Cd68*, *Tnf*, and *Hifa* were significantly downregulated in DBKO mice simultaneously lacking *ob* and *iNOS* genes as compared to those of *ob/ob* mice, revealing decreased liver inflammation (Figure 1a–d; Appendix A).

Compelling evidence has demonstrated a close link between liver inflammation, the production of ECM, and the development of fibrosis [29]. Therefore, the hepatic expression of the alarmin TNC was analyzed in the context of leptin and *iNOS* deficiency. As shown in Figure 1e, leptin-deficient *ob/ob* mice exhibited a significant increase in *Tnc* gene expression levels compared to wild-type mice. Moreover, serum TNC levels were significantly increased (*P* < 0.05) in *ob/ob* mice, and deletion of the *iNOS* gene reduced the circulating levels of this protein, confirming previous data described by our group (Figure 1f) [22].

Since TNC exhibits proinflammatory effects through the activation of TLR4, we compared the expression of genes involved in ECM remodeling in the absence of leptin and *iNOS*. Transcript levels of *Tlr4* were significantly (*P* < 0.05) increased in the liver of leptin-deficient mice, whereas the deletion of the *iNOS* gene in *ob/ob* mice dramatically reduced its expression (Figure 2a). Furthermore, gene expression levels of collagen type VI (*Col6a1* and *Col6a3*) as well as collagen type I (*Col1a1*) were upregulated in the absence of leptin, while their increased expression was reverted in DBKO mice counterparts (Figure 2b–d), indicating that iNOS is responsible for the collagen production to a certain extent, resulting in less collagen accumulation in *iNOS*-deficient mice. The gene expression levels of the gelatinase *Mmp9* were also markedly decreased (*P* < 0.01) in *iNOS* knockout and DBKO mice, whereas no differences where observed in *ob/ob* animals compared to wild-type mice (Figure 2e).

We next analyzed gene expression levels of osteopontin (*Spp1*), a multifunctional protein involved in liver diseases. *Spp1* mRNA significantly increased (*P* < 0.001) in leptin-deficient mice (Figure 2f). Moreover, gene expression levels of the osteopontin receptor CD44, recently described as an important marker and a key player of liver diseases, including NAFLD [30], were also determined. Leptin deficiency was associated with higher hepatic *Cd44* mRNA, while the deletion of *iNOS* significantly decreased (*P* < 0.05) its transcription levels (Figure 2g). 

Gene expression levels of the profibrogenic TGF-β were also analyzed. Deletion of *iNOS* significantly decreased (*P* < 0.01) *Tgfb* mRNA expression levels (Figure 2h). In order to elucidate the regulatory effect of the absence of the *ob* and *iNOS* genes in the activation of HSC, α-SMA (*Acta2*) gene expression levels were also determined. mRNA *Acta2* was dramatically downregulated (*P* < 0.01) after *iNOS* deletion (Figure 2i). Notably, a significant correlation between hepatic gene expression levels of *Tnc* and *Tlr4* (*r* = 0.442; *P* = 0.045) as well as with *Tnf* (*r* = 0.67; *P* = 0.001), *Col6a1* (*r* = 0.36; *P* = 0.048), *Col6a3* (*r* = 0.66; *P* = 0.001), *Col1a1* (*r* = 0.56; *P* = 0.008), *Acta2* (*r* = 0.68; *P* = 0.001), and *Tgfb* (*r* = 0.49; *P* = 0.023) was found.

### 3.3. Leptin Administration Protects from Inflammation and Fibrosis in the Liver of *ob/ob* Mice

The anthropometric and metabolic characteristics of *ob/ob* mice after leptin replacement or pair-feeding are reported in Appendix A. As expected, leptin administration ameliorated the obese and diabetic phenotype as well as improved lipid metabolism of *ob/ob* mice, corroborating previous findings of our group [22,31] and others [32,33]. Moreover, the increased liver weight observed in the absence of leptin was reversed (*P* < 0.001) by either caloric restriction or leptin replacement. The histological examination supported the results of the serum biochemical analysis: *ob/ob* liver sections exhibited macrovesicular steatosis that was completely reversed after leptin administration for 28 days, but not by caloric restriction (Figure 3a and Appendix A). 

The influence of leptin deficiency in liver fibrosis was next investigated. Analysis of Sirius Red-stained sections revealed that, in the control group of WT mice, staining was only observed in areas surrounding the blood vessels, whereas in *ob/ob* mice a slight intralobulillar liver fibrosis was observed. Liver fibrosis was less evident after leptin administration and caloric restriction (Figure 3a). Moreover, as shown in Figure 3b, *ob/ob* mice exhibited higher hepatic TG content, with the leptin administration and caloric restriction decreasing (*P* < 0.05) intrahepatic TG.

The inflammatory and fibrotic condition was also reversed in leptin-treated and pair-fed *ob/ob* mice, as evidenced by the decreased (*P* < 0.05) mRNA levels of factors related with the proinflammatory response, including *Emr1*, *Itgax*, *Tnfa*, *Hifa*, and *Tnc* (Figure 3c–g). Serum TNC levels were also normalized after leptin administration (Figure 3h). Moreover, it was also observed that leptin administration, as well as pair-feeding of *ob/ob* mice, prevented the increased mRNA expression of genes related to ECM remodeling such as *Tlr4*, *Col6a1*, *Col6a3*, *Col1a1*, *Mmp9*, and the marker of liver injury *Cd44*, the profibrogenic *Tgfb*, and the marker of activated of HSCs *Acta2* (Figure 4a–h). These data support the idea that the obese state is strongly associated with the inflammatory response and the extracellular matrix remodeling.

### 3.4. Leptin Treatment Increases Inflammatory and Fibrotic Genes in AML12 Hepatocytes

The direct effect of leptin treatment on the expression of inflammatory and fibrogenic genes was demonstrated in vitro using the non-tumorigenic mouse hepatocyte AML12 cell line. Upon 24 h leptin stimulation at physiological concentrations (10 nmol/L), *Hifa* mRNA expression was significantly upregulated (*P* < 0.05), and the gene expression levels of *Mmp9* and *Tlr4*, although differences were not statistically significant (*P* = 0.103 and *P* = 0.150, respectively) (Figure 5a–c). As expected, the expression of *Emr1* was not detected in AML12 hepatic cells (data not shown). Remarkably, leptin administration in AML12 hepatocytes significantly increased (*P* < 0.01) the expression of one of the most potent profibrogenic cytokine *Tgfb*, producing also a tendency towards an increased α-SMA (*Acta2*) gene expression (*P* = 0.150) (Figure 5d,e). Furthermore, leptin treatment upregulated the transcription of the *Tnc* gene (*P* = 0.09) with a significantly increased (*P* < 0.05) release of TNC (Figure 5f,g).

In order to determine the contribution of iNOS in mediating leptin-induced inflammation in AML12 hepatocytes, the effect of the pharmacological inhibition of iNOS with L-NIL, a selective iNOS inhibitor, was analyzed. Pharmacological inhibition of iNOS blunted the leptin-induced increase in *Tnc* mRNA after leptin stimulation as well as TNC release to the control media (Figure 5h,i). These results directly demonstrate the contribution of iNOS in *Tnc* expression in hepatic cells.

The stimulation with TNC significantly increased the expression levels of genes *Tlr4*, *Hif*, *Col6a1*, and *Col6a3* (*P* < 0.05) as well as *Tnf* (*P* = 0.08) in AML12 hepatic cells, without differences in *Mmp9* gene expression levels (data not shown) (Figure 6a–e), corroborating that this alarmin can induce a potent fibrogenic response. Noteworthy, TNC stimulation significantly increased the expression of both *Tgfb* and *Acta2* (*P* < 0.05) (Figure 6f,g).

## 4. Discussion

Adipose tissue is not only involved in energy storage but also functions as an endocrine organ secreting different bioactive adipokines [13,34]. The proinflammatory adipokine leptin regulates body weight and metabolism, exerting pleiotropic effects in many physiological systems including the liver, thereby linking obesity, insulin resistance, type 2 diabetes, and NAFLD [35,36,37]. Despite the wide range of reports regarding the participation of leptin in inflammation, its role in the pathogenesis of NAFLD remains unclear. However, increased serum leptin levels have been correlated with the amount of inflammation and fibrosis in liver diseases [38]. One of the key events in the promotion of liver diseases is the activation of HSCs [29]. Activated HSCs are the primary source of extracellular matrix components such as collagen or TNC, and accordingly fibrosis. Activated HSCs also express leptin, pointing to the implication of this factor during hepatic fibrogenesis and disease progression [11]. Moreover, our results also showed that leptin significantly and directly increased the proliferation of HSCs in a dose-dependent manner, confirming the ability of leptin to stimulate HSCs by the increased *Acta2* mRNA levels, a marker of HSC activation [39], in AML12 hepatic cells. Moreover, upon 24 h leptin stimulation, AML12 hepatocytes increased the expression of *Tgfb1*, considered a key driver in the activation of HSCs [40]. These results suggest that leptin, produced by HSC, increases *Tgfb1* expression, which, in turn, stimulates fibrogenesis in HSC in an intricate paracrine loop. Furthermore, the pro-inflammatory capacity of leptin promotes or sustains low-grade inflammation [41,42]. In this context, leptin administration increases both inflammation and fibrogenesis in AML12 hepatocytes. Leptin also stimulated the synthesis and release of TNC in the liver, which interacts with several ECM proteins and cell receptors, including TLR4. TLRs are not only important in the regulation of innate and adaptive immune responses, but they are also involved in inflammatory diseases of the cardiovascular system and liver [43]. The TNC/TLR4 signaling axis is fundamental for the induction of proinflammatory cytokines and the ECM remodeling, among other functions [44]. In this regard, our group has recently reported that TNC, through TLR4, is implicated in the etiopathology of obesity adipose tissue inflammation [22]. Several studies support that TNC is involved in liver fibrosis, in part due to its distribution in areas of lymphocytic infiltration, contributing to liver fibrogenesis through the enhancement of the inflammatory response, the promotion of HSC activation, and the enhancement of TGF-β expression [9]. In order to determine if TNC induces potent fibrotic and inflammatory responses not only in fibroblasts [45] but also in hepatocytes, AML12 cells were stimulated with this alarmin. The stimulation of murine AML12 hepatocytes with TNC increased the expression of *Tlr4*, *Hif1a*, *Tnfa*, and the fibrogenic genes *Col6a1* and *Col6a3*, confirming the important role of TNC in both liver fibrogenesis by increasing the synthesis of collagen and in the hepatic inflammatory response.

Converging lines of evidence have demonstrated that leptin exerts a regulatory role in the connection between energy metabolism and the immune system, being a crucial adipokine responsible for the inflammatory state found in obesity [46]. Leptin-deficient *ob/ob* mice exhibited increased liver inflammatory and fibrotic response, as evidenced by the increased infiltration of liver macrophages, collagen production, and fibrotic matrix deposition. In this context, hypoxia is considered another key microenvironmental factor contributing to inflammation and fibrosis in liver diseases. *Hif1a* can be activated by different mediators in addition to hypoxia, including pro-inflammatory cytokines or oxidative stress, and regulates the activation of the profibrogenic factor TGF-β, promoting fibrogenesis. In line with these observations, our results showed that genes related to the regulation of hypoxic response (*Hif1a*), inflammation (*Tnfa*, *Itgax*, *Cd68*), and fibrogenesis (*Col6a1*, *Col6a3*, *Col1a1*) were highly upregulated in the liver of leptin-deficient mice. Furthermore, circulating levels as well as liver expression of TNC were also highly increased in *ob/ob* mice, contributing to inflammation, to the activation of matrix-producing cells, and to matrix deposition and remodeling associated to obesity. According to previous studies from our group, the obese and diabetic phenotype of *ob/ob* mice, as well as the elevated expression of hypoxic, proinflammatory, and profibrotic genes were restored after chronic leptin administration, and may contribute, together with the reduction in lipogenesis and the increase in fatty-acid oxidation, to an improvement of hepatic function. These data support that leptin administration in *ob/ob* mice reverses the conditions of inflammation, fibrogenesis, and extracellular matrix remodeling, normalizing the metabolic status and exerting anti-steatotic effects in the liver. Nonetheless, our data suggest that these phenomena are not exclusively produced by leptin, since similar results were obtained in the pair-fed group, suggesting that other factors might be involved in the beneficial effect observed in the inflammatory and fibrotic response independently of weight loss [31,47]. In this sense, increased oxidative stress and elevated systemic inflammation constitute a general phenomenon of the obese state, also observed in the context of leptin deficiency [23,48]. The increased levels of pro-inflammatory cytokines observed in *ob/ob* mice may interact with HSCs, inducing collagen gene deposition and hepatic fibrogenesis. Leptin potentiates fibrosis but does not constitute an essential factor for fibrogenesis in the liver.

Excessive fat accumulation in the liver augments ROS formation, inducing the expression of pro-inflammatory genes including TNF-α, IL-6, and cyclooxygenase-2. This, in turn, induces the expression of additional inflammatory mediators that interact with HSCs, increasing the profibrotic response [49]. Although under normal physiological conditions NO is generated constitutively and iNOS expression is absent, different disease conditions, including hepatic fibrosis, induce its expression, mainly in Kupffer cells and HSCs. The excessive NO generation triggers key processes involved in NAFLD progression, including mitochondrial biogenesis and function [50], Kupffer cell polarization [51], and HSC fibrogenesis [52], demonstrating an important role of iNOS in liver inflammation and fibrogenic response [53]. In line with this, *iNOS* deletion significantly reduced the activation of HSCs as confirmed by the reduced gene expression levels of both *Tgfb* and a-SMA (*Acta2*), suggesting that *iNOS* deficiency could regress the activation of HSC by interfering with the TGF-β pathway, protecting from liver fibrosis.

Given that many biological actions of leptin are mediated by NO [15,16,18,20,21,22], we aimed to evaluate if a functional relationship among them in liver inflammation and fibrosis in the context of obesity. For that purpose, we examined the effects of *iNOS* gene disruption in genetically *ob/ob* mice. We provide evidence, for the first time, that iNOS is involved in liver inflammation and fibrosis linked to leptin deficiency. Several evidences support the improved metabolic profile as well as liver inflammation and fibrosis in the DBKO mice. First of all, a causal relationship between excess fat accumulation, insulin resistance, and progression of hepatic fibrosis is well established [54]. The improved metabolic profile observed in DBKO mice, in line with previous work of our group [22], may constitute an important cornerstone of liver improvement: the decreased accumulation of lipids in hepatocytes together with the increased levels of adiponectin and the reduced liver inflammatory profile in DBKO mice is in accordance with the improved obese and diabetic phenotype of our DBKO model [22].

Moreover, we found that genes involved in inflammation, in the regulation of hypoxic response, and in the excessive collagen deposition are highly enriched in livers of *ob/ob* mice, whereas this upregulation was completely reverted by *iNOS* deletion, similar to previous results described in adipose tissue [22]. Of note, hypoxia induces and regulates iNOS expression, and NO produced by iNOS participates in the stability control of HIF-1α [55]. In line with these observations, *iNOS* deletion, via decreased transcription and stability of HIF-1α, might improve liver hypoxia in *ob/ob* mice.

Another important finding of the present study is that the upregulation of osteopontin (*Spp1*) and its receptor *Cd44* observed in *ob/ob* mice was completely prevented by iNOS inhibition. The transmembrane protein CD44 has been recently described as a marker and key player of NASH development [30] and one of the main osteopontin (OPN) receptors [56]. OPN expression is upregulated by proinflammatory cytokines including TNF-α and TGF-β and, among other functions, plays a major role in the adipose tissue expansion and in the development of liver statosis [57,58,59]. It is reported that NO enhances the expression of OPN in the context of hepatic carcinoma associated with high levels of iNOS expression [60]. Our group has previously reported that the absence of OPN reversed HFD-induced fatty liver [57], suggesting that OPN and its corresponding receptor CD44 play a critical role in liver fibrosis, as both are decreased in the absence of the *iNOS* gene. *iNOS* deletion is also related with a profound downregulation of genes encoding collagen, a main structural component of ECM, including *Col61a1*, *Col6a1*, and *Col613*. Besides the regulatory function of NO on Mmp9 activation [61], the present study demonstrates that *Mmp9* gene expression levels were also significantly reduced in the absence of *iNOS*. 

It is necessary to emphasize that *iNOS* ablation significantly reduces circulating levels and liver expression of TNC and its potential role in the stimulation of proinflammatory cytokine expression. Moreover, iNOS inhibition also induced a downregulation of basal and leptin-induced *Tnc* transcript levels, suggesting that leptin induces TNC via the activation of the iNOS enzyme, with leptin-induced TNC upregulation preventing *iNOS* inhibition.

Collectively, the present study identified a novel relationship for iNOS and leptin in mediating liver fibrogenesis. iNOS deficiency improves liver inflammation as well as the expression of ECM remodeling-related genes in the context of leptin deficiency, attenuating the development of fibrosis. The weakening of the ECM due to ablation of the *iNOS* gene could be associated with improved insulin sensitivity, reduced inflammation, and metabolic benefits. Moreover, iNOS activation induced by leptin is needed and crucial for the synthesis and release of the profibrogenic and proinflammatory TNC, suggesting an important role of this alarmin in the development of hepatic inflammation and fibrosis (Figure 6h). Leptin, together with other important factors involved in increased oxidative stress and elevated systemic inflammation, may contribute to liver steatosis in *ob/ob* mice. Thus, the participation of other factors altered by obesity that impinge on both adipose tissue and liver like fibroblast growth factors (FGFs) should be considered [62]. Further research is needed to discern the specific role of leptin and iNOS in liver fibrogenesis.

## Figures and Tables

**Figure 1 genes-10-00184-f001:**
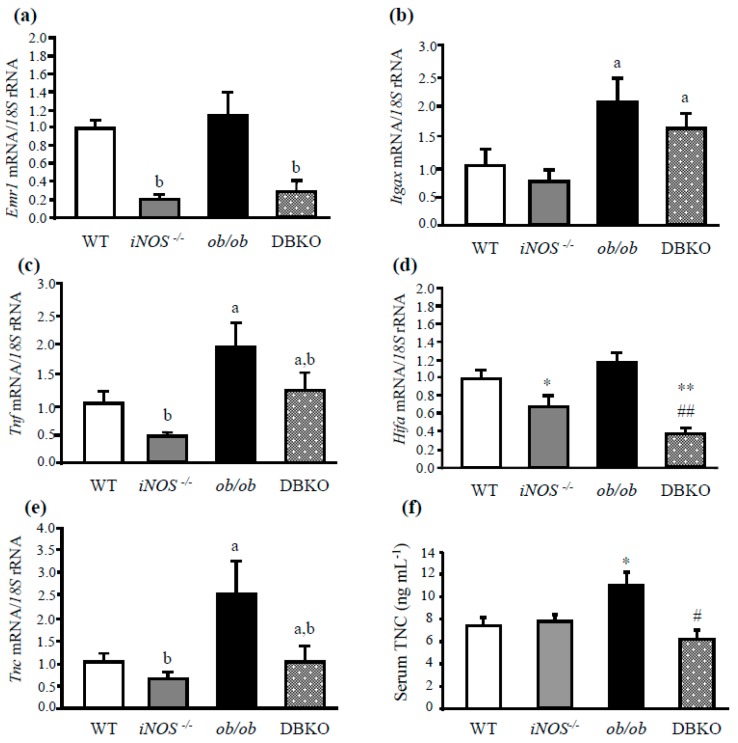
Hepatic expression of genes involved in inflammation and fibrosis. Gene expression levels of proinflammatory markers *Emr1* (**a**), *CD11c (Itgax)* (**b**), *Tnf* (**c**), *Hif1a* (**d**), and tenascin C (*Tnc*) (**e**) in the liver (*n* = 5–6). The gene expression in wild type (WT) mice was assumed to be 1. Serum TNC levels (**f**) of the different experimental groups (*n* = 5 per group). Differences between groups were analyzed by two-way ANOVA or one-way ANOVA followed by Tukey’s post hoc test when an interaction between factors was detected. ^a^*P* < 0.05 effect of the absence of the *ob* gene. ^b^*P* < 0.05 effect of the absence of the *iNOS* gene. **P* < 0.05, ***P* < 0.01 *vs* WT mice; #*P* < 0.05, ##*P* < 0.01 *vs ob/ob* mice. iNOS: Inducible nitric oxide synthase; DBKO: double knockout.

**Figure 2 genes-10-00184-f002:**
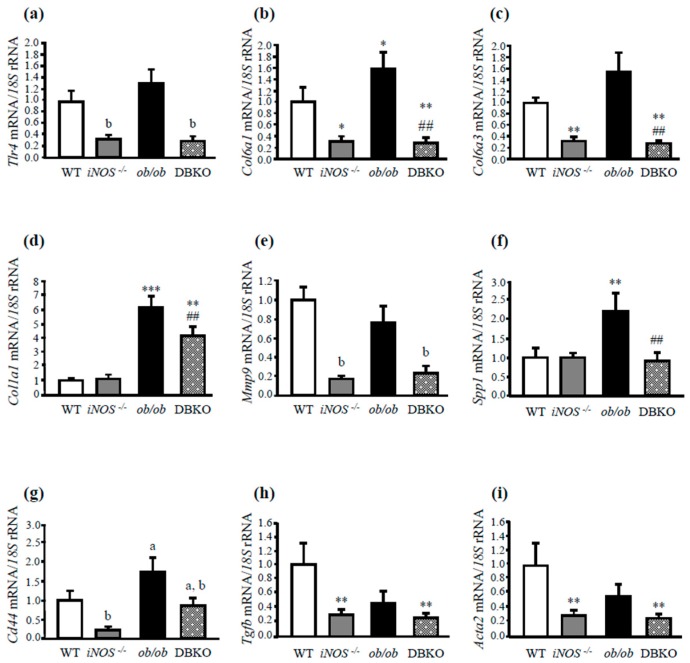
Effect of the absence of the *iNOS* gene in liver fibrosis in the context of leptin deficiency. Gene expression levels of *Tlr4* (**a**), *Col6a1* (**b**), *Col6a3* (**c**), *Col1a1* (**d**), *Mmp9* (**e**), *Spp1* (**f**), and *Cd44* (**g**) in the liver. Gene expression levels of liver *Tgfb* (**h**) and α-SMA *(Acta2)* (**i**) (n = 5) were also analyzed. The gene expression in WT mice was assumed to be 1. Differences between groups were analyzed by two-way ANOVA or one-way ANOVA followed by Tukey’s post hoc test when an interaction between factors was detected. ^a^*P* < 0.05 effect of the absence of the *ob* gene. ^b^*P* < 0.05 effect of the absence of the *iNOS* gene. **P* < 0.05, ***P* < 0.01, ****P* < 0.001 *vs* WT mice; ##*P* < 0.01 *vs ob/ob* mice.

**Figure 3 genes-10-00184-f003:**
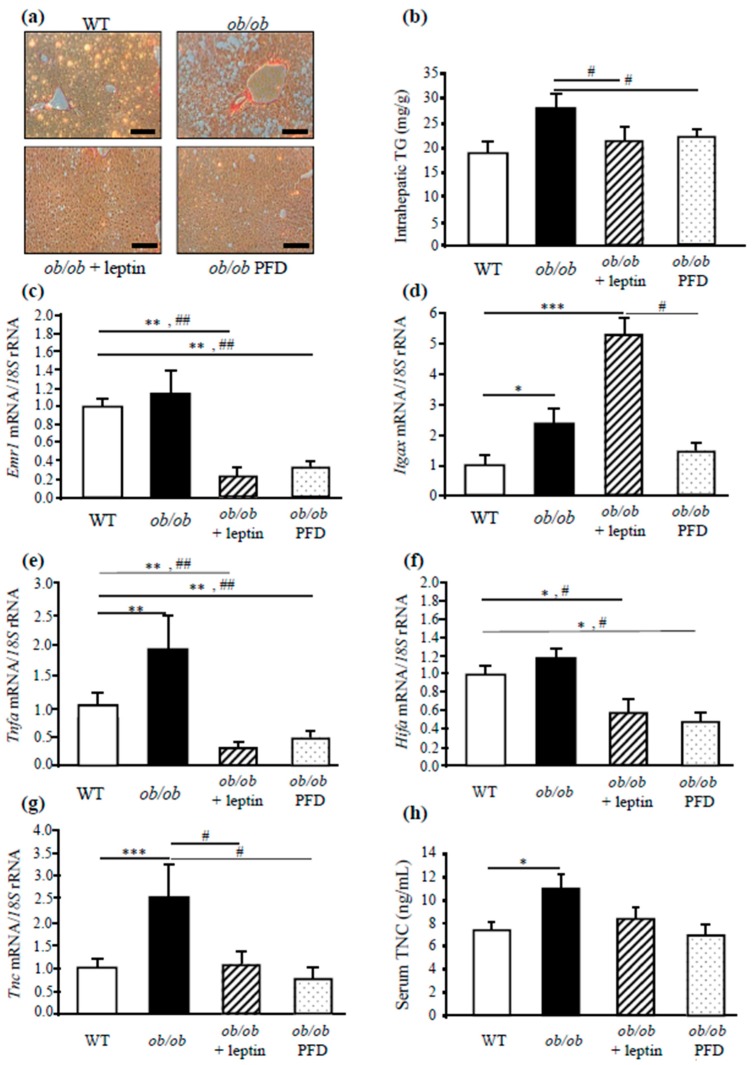
Effect of in vivo leptin administration on the liver phenotype and expression of genes involved in hepatic inflammation in *ob/ob* mice. Representative Sirius Red staining (magnification 200×, scale bar = 50 µm, n = 3/per group) (**a**) and intrahepatic TG content (n = 4–5 per group) (**b**). Gene expression levels of *Emr1* (**c**), *CD11c (Itgax)* (**d**), *Tnfa* (**e**), *Hif1a* (**f**), and *Tnc* (**g**) in the liver (n = 5–6) in wild-type (WT), *ob/ob*, leptin-treated, and pair-fed *ob/ob* mice (n = 5–6 per group). Gene expression levels in WT mice in liver were assumed to be 1. Serum TNC levels (**h**) of the different experimental groups (n = 5 per group). Comparisons between *ob/ob* groups and controls were analyzed by one-way ANOVA followed by Tukey’s post hoc tests. **P* < 0.05, ***P* < 0.01, ****P* < 0.001 *vs* WT mice; #*P* < 0.05, ##*P* < 0.01 *vs ob/ob* mice. PFD: Pair-fed group.

**Figure 4 genes-10-00184-f004:**
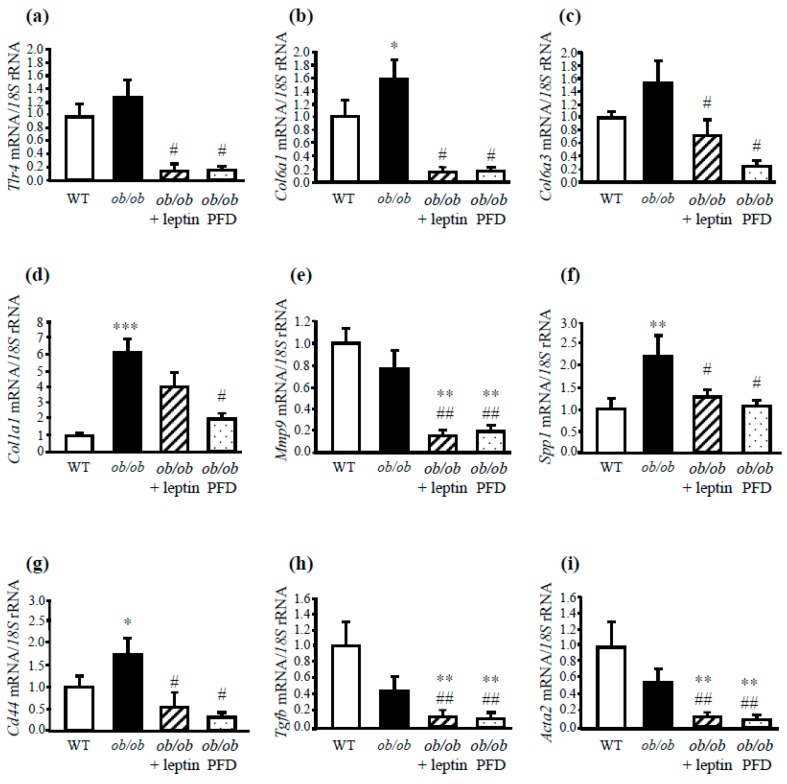
Effect of leptin replacement on the expression of genes involved in liver fibrosis in *ob/ob* mice. Gene expression levels of *Tlr4* (**a**), *Col6a1* (**b**), *Col6a3* (**c**), *Col1a1* (**d**), *Mmp9* (**e**), *Spp1* (**f**), and *Cd44* (**g**) in the liver of WT, *ob/ob*, leptin-treated, and pair-fed *ob/ob* mice (n = 5–6 per group). Gene expression levels of *Tgfb* (**h**) and α-SMA (*Acta2*) (**i**) were also evaluated. Gene expression levels in WT mice were assumed to be 1. Comparisons between *ob/ob* groups and controls were analyzed by one-way ANOVA followed by Tukey’s post hoc tests. **P* < 0.05, ***P* < 0.01, ****P* < 0.001 *vs* WT mice; #*P* < 0.05, ##*P* < 0.01 *vs ob/ob* mice.

**Figure 5 genes-10-00184-f005:**
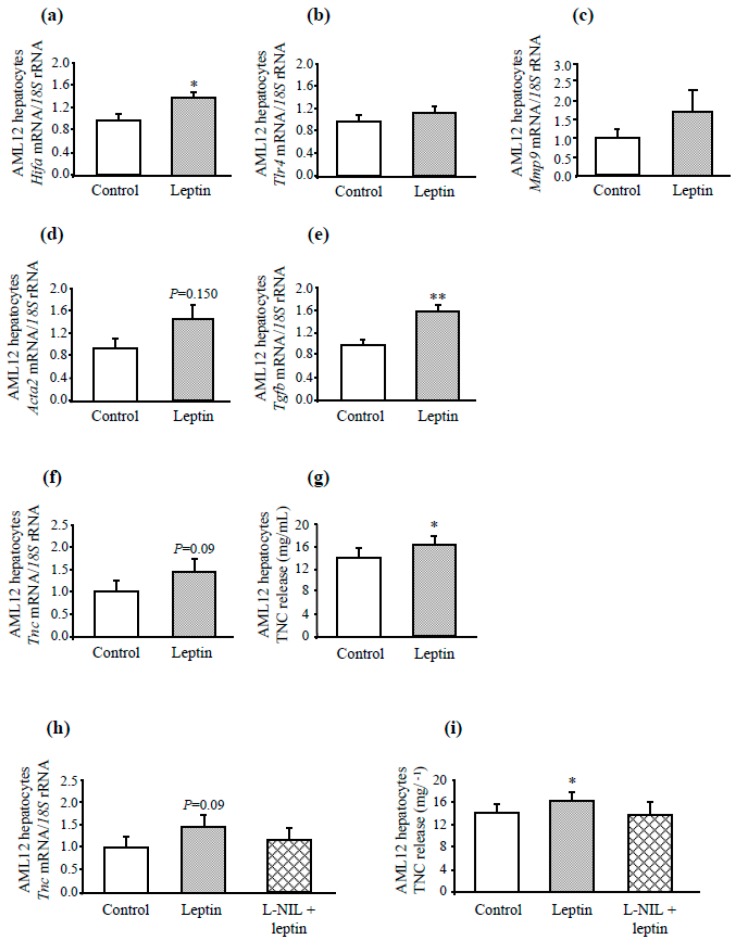
Effect of leptin stimulation for 24 h on the expression of markers of inflammation and fibrosis in AML12 hepatocytes. Effect of leptin stimulation (10 nmol L^−1^) for 24 h on gene expression levels of *Hif1a* (**a**), *Tlr4* (**b**), *Mmp9* (**c**), α-SMA (*Acta2*) (**d**), *Tgfb* (**e**), and *Tnc* (**f**) as well as TNC release to the culture media (**g**) in AML12 liver cells (n = 6 per group). *Tnc* transcript levels (**h**) and TNC release (**i**) to the culture media in AML12 adipocytes stimulated with leptin (10 nmol/L) in the absence or presence of the iNOS inhibitor l-NIL (10 mmol/L) for 24 h. Gene expression levels in the unstimulated cells were assumed to be 1. Values are the mean ± SEM (n = 5 per group). Differences between groups were analyzed by unpaired two-tailed Student’s *t*-tests. **P* < 0.05; ***P* < 0.01 *vs* unstimulated cells.

**Figure 6 genes-10-00184-f006:**
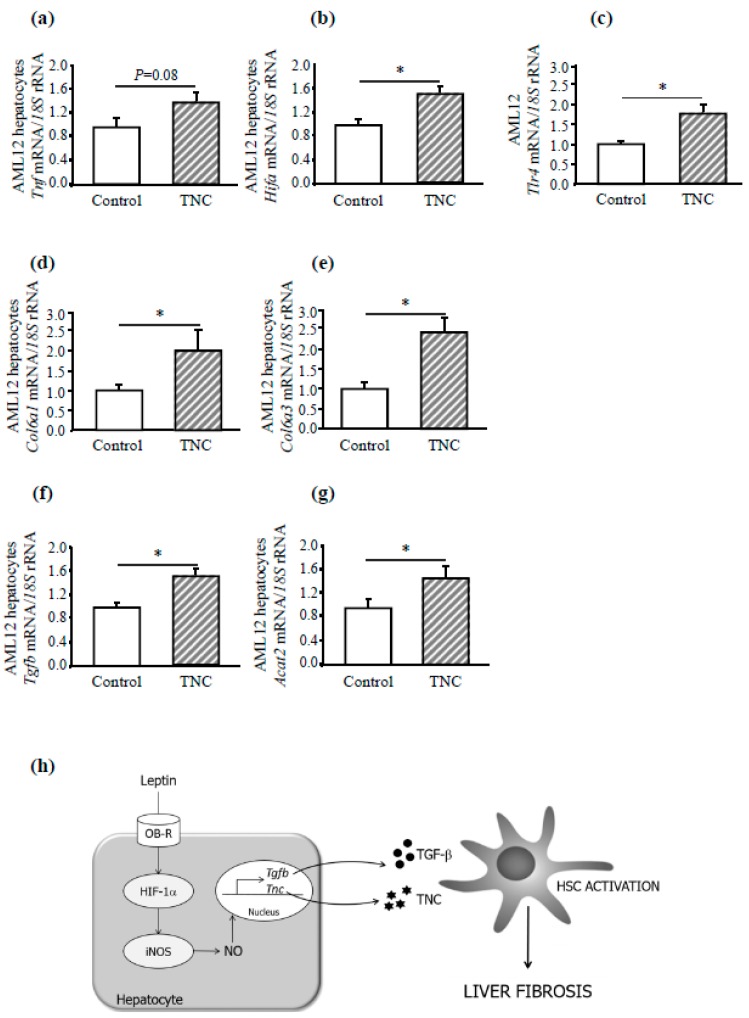
Effect of tenascin C (TNC) stimulation for 24 h on the expression of markers of inflammation and fibrosis in AML12 hepatocytes. Effect of TNC stimulation (10 nmol/L) for 24 h on gene expression levels of *Tnfa* (**a**), *Hif1a* (**b**), *Tlr4* (**c**), *Col6a1* (**d**), *Col6a3* (**e**), as well as *Tgfb* (**f**) and *a-SMA (Acta2)* (**g**) in AML12 hepatocytes (*n* = 5 per group). Differences between groups were analyzed by unpaired two-tailed Student’s *t*-tests. **P <* 0.05 *vs* unstimulated cells. Summary graph for the regulation of liver fibrosis induced by leptin through iNOS activation (**h**).

**Table 1 genes-10-00184-t001:** Anthropometric and metabolic characteristics of 12-week-old experimental animals.

	Wild Type	*iNOS* ^-/-^	*ob/ob*	DBKO
Body weight (g)	24.0 ± 0.4	23.1 ± 0.4	44.9 ± 1.5	43.1 ± 0.8
Liver weight (g)	1.04 ± 0.02	1.00 ± 0.03	3.15 ± 0.12	2.97 ± 0.11
FFA (mmol/L)	0.74 ± 0.08	0.58 ± 0.03	1.07 ± 0.06^***^	0.85 ± 0.06
TG (mg/dL)	66.3 ± 4.4	77.1 ± 3.9	97.3 ±5.3^**^	90.4 ± 0.1 ^†^
Glycerol (mg/dL)	0.025 ± 0.001	0.022 ± 0.003	0.036 ±.0.001^***^	0.030 ± 0.003^***, ††^
Intrahepatic TG (mg/g)	19.0 ± 2.2	16.8 ± 1.5	28.5 ± 3.0	24.5 ± 2.6
Glucose (mg/dL)	83 ± 5	77± 2	410 ±42^***^	96 ± 5^***, ††^
Insulin (ng/mL)	0.42 ± 0.04	0.32 ± 0.04	9.66 ± 0.61^***^	8.89 ± 1.17^***,†††^
HOMA	1.6 ± 0.2	1.1 ± 0.2	172.3.0 ± 21.5^***^	34.5 ± 5.2^***, ††^
Adiponectin (μg/mL)	22.6 ± 4.1	25.6 ± 3.1	16.6 ± 0.9^***^	20.1 ± 5.0^**,††^

BW: body weight; DBKO: double knockout mice simultaneously lacking *ob* and *iNOS* genes; FFA: free fatty acids; HOMA: homeostasis model assessment; iNOS: Inducible nitric oxide synthase. TG: triacylglycerols. Data are mean ± SEM (n = 4–5 per group). Differences between groups were analyzed by two-way ANOVA or one-way ANOVA followed by Tukey’s post hoc test when an interaction between factors was detected. ^*^*P* < 0.05, ^**^*P* < 0.01, ^***^*P* < 0.001 *vs* wild type mice; ^†^
*P* < 0.05, ^††^*P* < 0.01 *vs ob/ob* mice.

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
