# Peer review of "iNOS Gene Ablation Prevents Liver Fibrosis in Leptin-Deficient ob/ob Mice"

_genes, 2019, doi:10.3390/genes10030184_

Round 1
Reviewer 1 Report
The manuscript by Sara Becerril et al. is interesting. The experiments are well designed and data are novel and sound.
Nevertheless, I have a major concern regarding the conclusions raised by the present work. On the one hand the obese model employed is deficient in leptin (Ob mice), developing a fatty liver, due to inflammatory signals, and mediated by iNOS, sine iNOS ablation prevents the liver steatosis. But on the other hand, the authors have found that leptin induces the expression of iNOS, and they state that leptin is necessary for its effect on liver inflammation and fibrosis. Thus, the authors conclude: "leptin is needed for the synthesis and release of the profibrogenic and proinflammatory TNC, suggesting an important role of his alarmin in the development of hepatic inflammation and fibrosis"
The obese model employed demonstrate that leptin is not necessary for the development of fatty liver in obesity. Instead, the conclusion should be the opposite. iNOS expression seems to be crucial for the liver inflammation and fibrosis, but it seems clear from these data that not leptin, but other inflammatory signals may contribute to the liver steatosis in ob-/ob- mice.
In any case, data regarding the role of leptin on iNOS expression demosntrate its possible role on liver fibrosis, along with other inflammatory signals in obesity
Author Response
iNOS gene ablation prevents liver fibrosis in leptin-deficient ob/ob mice
We would like to thank the Reviewer for his/her kind and thoughtful comments, which
have been very encouraging.
Specific comments:
The manuscript by Sara Becerril et al. is interesting. The experiments are well
designed and data are novel and sound.
Nevertheless, I have a major concern regarding the conclusions raised by the
present work. On the one hand, the obese model employed is deficient in leptin (Ob
mice), developing a fatty liver, due to inflammatory signals, and mediated by
iNOS, since iNOS ablation prevents the liver steatosis. On the other hand, the
authors have found that leptin induces the expression of iNOS, and they state that
leptin is necessary for its effect on liver inflammation and fibrosis. Thus, the
authors conclude: "leptin is needed for the synthesis and release of the
profibrogenic and proinflammatory TNC, suggesting an important role of his
alarmin in the development of hepatic inflammation and fibrosis".
The obese model employed demonstrate that leptin is not necessary for the
development of fatty liver in obesity. Instead, the conclusion should be the
opposite. iNOS expression seems to be crucial for the liver inflammation and
fibrosis, but it seems clear from these data that not leptin, but other inflammatory
signals may contribute to the liver steatosis in ob/ob mice.
In any case, data regarding the role of leptin on iNOS expression demonstrate its
possible role on liver fibrosis, along with other inflammatory signals in obesity
We thank the Reviewer for the positive evaluation of our study, and for the excellent
comments that have helped to further improve the manuscript.
The Reviewer is right in his/her appreciation. One important point of this study is that
iNOS activation induced by leptin is needed for the synthesis and release of the
profibrogenic and proinflammatory factors. Moreover, the obese state, strongly
associated with the inflammatory response and the extracellular matrix remodelling, is
not exclusively caused by leptin, but also by other important factors involved in the
increased oxidative stress and the elevated systemic inflammation associated with
obesity. Thus, the participation of other factors altered by obesity that impinge on
both adipose tissue and liver like fibroblast growth factors (FGFs) should be
considered. The Reviewer’s comment has been incorporated to the Discussion section
of the revised version of the manuscript and the conclusion has been changed
accordingly.
Gallego-Escuredo JM, Gómez-Ambrosi J, Catalan V, Domingo P, Giralt M, Frühbeck G, Villarroya F. Opposite alterations in FGF21 and FGF19 levels and disturbed expression of the receptor machinery for endocrine FGFs in obese patients. Int J Obes (Lond). 2015 Jan;39(1):121-9.
Reviewer 2 Report
By comparing the phenotype of Leptin deficient (ob/ob) and iNOS-deficient mice, Sara Becerril and his coworkers found that “iNOS deficiency in ob/ob mice improved liver inflammation and ECM remodelling-related genes, decreasing fibrosis and metabolic dysfunction”. After carefully reading, I have following concerns:
Major:
1, Since iNOS deficiency alone could lead to increased inflammation and fibrosis. It’s interesting to inquire if iNOS is independent of leptin, or they have an interactional or even causal relationship. So, it will be interesting if the author could show us the NO level in ob/ob and wildtype mice.
2, To demonstrate the increased inflammation, they author use qPCR to show us the up-regulated of the Itgax mRNA, this is not enough to demonstrate the distribution of infiltrating macrophages. The author should also perform immunofluorescent staining assay to show the pattern of F4/80 or cd11b.
3, The function of leptin here is not clear. The author showed in result 3.3 that “Leptin administration protects from inflammation and fibrosis in the liver of ob/ob mice”, while in result 3.4 that “Leptin treatment increases inflammatory and fibrotic genes in AML12 hepatocytes”. These results seem paradoxical and confusing. We don’t know if leptin can suppress or promote inflammation. The author should discuss and clarify this more.
Minor:
1, The background of Fig. 3A is too strong, which need to be repeated.
2, The significance of qPCR result is not clearly demonstrated in the figure, the author should draw a line to connect the control and the examined group and give the significant symbol above the line.
2, Some word showed different size, such as line 294. Also, we can see a light shadow in line 352. Moreover, we observed a lot of unknown symbols in the manuscript and even a bird in line 155.
Author Response
iNOS gene ablation prevents liver fibrosis in leptin-deficient ob/ob mice
We are very grateful for the detailed comments of the Reviewer, which have served to
substantially improve the manuscript.
Major:
1. Since iNOS deficiency alone could lead to increased inflammation and fibrosis.
It’s interesting to inquire if iNOS is independent of leptin, or they have an
interactional or even causal relationship. So, it will be interesting if the author
could show us the NO level in ob/ob and wild type mice.
We greatly appreciate the comment of the Reviewer regarding the analysis of serum
NOx concentration in ob/ob and wild type mice. Unfortunately, we do not have sera
samples affordable to perform this determination. However, iNOS is known to be
upregulated under conditions related to acute inflammation, with NO being a key
messenger in the pathogenesis of inflammation. In fact, iNOS expression is significantly
elevated in the liver as well as skeletal muscle and adipose tissue in ob/ob mice
compared with wild-type mice (Fujimoto et al, 2005), suggesting that serum NOx
concentrations is elevated in ob/ob mice as compared with wild type littermates.
Elizalde M, Rydén M, van Harmelen V, Eneroth P, Gyllenhammar H, Holm C, Ramel S, Ölund A, Arner P, Andersson K: Expression of nitric oxide synthases in subcutaneous adipose tissue of nonobese and obese humans. J Lipid Res41:1244–1251, 2000
Fujimoto M, Shimizu N, Kunii K, Martyn JA, Ueki K, Kaneki M. A role for iNOS in fasting hyperglycemia and impaired insulin signaling in the liver of obese diabetic mice. Diabetes. 2005 May; 54(5):1340-8.
2. To demonstrate the increased inflammation, they author use qPCR to show us
the up-regulated of the Itgax mRNA, this is not enough to demonstrate the
distribution of infiltrating macrophages. The author should also perform
immunofluorescent staining assay to show the pattern of F4/80 or cd11b.
The Reviewer raises a very interesting point, which is the analysis of the protein levels
by immunofluorescent staining in addition to Real-time PCR. Unfortunately, we have
no remaining liver sections from the mice included in the expression analysis for the
mRNA studies. However, this is a very interesting suggestion that will be taken into
consideration for future studies. However, following the Reviewer’s suggestion, we
have confirmed the presence of infiltrating macrophages through the analysis of gene
expression levels of Cd68, another important marker for resident macrophages together
with Itgax and Emr1 (Antoniades et al, 2012). As expected, leptin-deficient ob/ob mice
exhibited a significant increase in Cd68 mRNA levels compared to wild type mice, with
iNOS disruption reducing Cd68 transcript levels. The new figure (Supplemental Fig. 1)
and the reference have been included in the revised version of the manuscript.
3. The function of leptin here is not clear. The author showed in result 3.3 that “Leptin administration protects from inflammation and fibrosis in the liver of ob/ob mice”, while in result 3.4 that “Leptin treatment increases inflammatory and fibrotic genes in AML12 hepatocytes”. These results seem paradoxical and confusing. We don’t know if leptin can suppress or promote inflammation. The author should discuss and clarify this more.
We certainly agree with the Reviewer that the results may seem paradoxical and confusing. On one hand, leptin-deficient ob/ob mice are obese, hyperphagic, exhibit type 2 diabetes, defective body temperature and hypogonadism. These metabolic abnormalities can all be reversed by leptin administration (Sainz et al, 2010; Halaas et al, 1995; Pelleymounter et al, 1995; Chehab, 1996). In the present study it has been observed that, in the context of liver inflammation and fibrosis, leptin treatment restores these conditions by decreasing mRNA levels of factors related with inflammation and extracellular matrix remodelling, suggesting that leptin normalizes the metabolic status in the liver ob ob/ob mice. On the other hand, leptin has pro-inflammatory properties and several actions similar to those of the acute phase reactants, upregulating the secretion of inflammatory cytokines including TNF-α, IL-6, or IL-12. This pro-inflammatory capacity promotes or sustains low.grade inflammation, favouring the development of chronic diseases (Shen et al, 2005; Faggioni et al, 2000). In this context, leptin administration increases inflammation and fibrogenesis in AMLS12 hepatocytes.
These comments have been included in the Discussion section and references have been added to the revised version of the manuscript.
Minor:
1. The background of Fig. 3A is too strong, which need to be repeated.
Following the Reviewer's suggestion, and in order to improve the quality of the images, we have performed a new captioin of the liver sections of the experimental groups, and a new figure has been included in the revised version of the manuscript (Fig. 3a).
2. The significance of qPCR result is not clearly demonstrated in the figure, the author should draw a line to connect the control and the examined group and give the significant symbol above the line.
According to Reviewer's suggestion, a line connecting the control and the examined groups has been included in the Results' section (Figs. 1-6) of the revised version of the manuscript.
Reviewer 3 Report
The role of extracellular matrix (ECM) remodeling in fibrosis progression in nonalcoholic fatty liver disease (NAFLD) is complex and dynamic, involving the synthesis and degradation of different ECM components, including tenascin C (TNC). It’s a very good idea to look at the influence of iNOS deletion on inflammation and ECM remodeling in the liver of ob/ob mice. This study demonstrated that the activation of iNOS by leptin is necessary for the synthesis and secretion of TNC in hepatocytes. However, more detail investigations are need as well.
1. There are many written mistakes, would be nice to correct them
2. Most of the data are associated with mRNA expression level, would be great to show the changes in protein expressing level
3. It would be very useful to have a summary graph for the regulation and signal pathway
4. It would be very useful to show more staining images for confirmation.
Author Response
iNOS gene ablation prevents liver fibrosis in leptin-deficient ob/ob mice
We would like to thank the Reviewer for his/her kind and thoughtful comments, which have been very encouraging.
1. There are many written mistakes, would be nice to correct them.
They typo and grammar errors mentioned by the Reviewer have been amended and the whole manuscript has been carefully revised.
2. Most of the data are associated with mRNA expression level, would be great to show the changes in protein expression levels.
The Reviewer raises a very interesting point, shich is the analysis of the protein levels by Western blot in addition to real-time PCR. Unfortunately, we have no remaining liver samples from the mice included in the expression analysis for the mRNA studies. However this is a very interesting suggestion that will be taken into consideration for future studies.
3. It would be useful to have a summary graph for the regulation and signal pathway.
According to Reviewer's suggestion, we have included a summary graph for the regulation of liver fibrosis induced by leptin through iNOS activation (Fig. 6h).
4. It would be useful to show more staining images for confirmation.
The point raised by the reviewer is interesting and very similar to point 2. It would be very interesting to show more staining images for confirmation. Unfortunately, we have no additional samples available to perform more immunofluorescent staining assays.
Round 2
Reviewer 1 Report
The authors have answered my questions, and I think it may be published.
Reviewer 2 Report
I think it fits publication now.